# The origin of magnetization-caused increment in water oxidation

Xiao Ren[1,2,3,7], Tianze Wu [1,7], Zizhao Gong[1,2], Lulu Pan[2], Jianling Meng[4], Haitao Yang [2], Freyja Bjork Dagbjartsdottir[5], Adrian Fisher[5], Hong-Jun Gao [2] & Zhichuan J. Xu [1,6]

Magnetization promoted activity of magnetic catalysts towards the oxygen evolution reaction (OER) has attracted great attention, but remains a puzzle where the increment comes from. Magnetization of a ferromagnetic material only changes its magnetic domain structure. It does not directly change the spin orientation of unpaired electrons in the material. The confusion is that each magnetic domain is a small magnet and theoretically the spin-polarization promoted OER already occurs on these magnetic domains, and thus the enhancement should have been achieved without magnetization. Here, we demonstrate that the enhancement comes from the disappeared domain wall upon magnetization. Magnetization leads to the evolution of the magnetic domain structure, from a multi-domain one to a single domain one, in which the domain wall disappears. The surface occupied by the domain wall is reformatted into one by a single domain, on which the OER follows the spin-facilitated pathways and thus the overall increment on the electrode occurs. This study fills the missing gap for understanding the spin-polarized OER and it further explains the type of ferromagnetic catalysts which can give increment by magnetization.

Toward a carbon-zero energy infrastructure, it is critical to implement the techniques such as water electrolysis powered by sustainable energy resources for green hydrogen generation. However, the electrolysis efficiency is greatly limited by the sluggish anode kinetics in water electrolyzers, i.e., the oxygen evolution reaction (OER)[1–3]. Recently, it has been reported that OER on some magnetic catalysts can be promoted by applying an external magnetic field[4–6]. Such a phenomenon has attracted great interest crossing the fields of energy, materials, chemistry, and physics, and it is also under debate. Most recently, it has been confirmed by excluding other possible contributing factors, including Lorenz force-promoted mass transportation[5], lowered resistance by magnetization[6], and magnetic flux-induced bias[7]. On the other hand, the earlier theoretical study by Jose et al. has predicted that the spin-polarized electron transfer is a prerequisite for the fast generation of triplet oxygen from singlet reactants ($H_2O$ and $OH^-$)[8,9] and it can be facilitated on the surface of ferromagnetic catalysts where the open-shell spin interactions reduce the barrier of electron transfer[10,11]. The most recent progress by us has revealed the critical role of M-O radical species in magnetization-promoted OER[12], which is concluded from the fact that the increment is pH-dependent. However, it should be noted that the magnetization of catalysts has no influence on the spin ordering of unpaired electrons in atoms in ferromagnetic materials, including the metal active sites[13]. Magnetization only leads to the evolution of magnetic structure in a

[1]School of Material Science and Engineering, Nanyang Technological University, 50 Nanyang Avenue, Singapore 639798, Singapore. [2]Beijing National Laboratory for Condensed Matter Physics and Institute of Physics, Chinese Academy of Science, P.O.Box 603 Beijing 100190, China. [3]Beijing National Laboratory for Molecular Engineering, College of Chemistry and Molecular Engineering, Peking University, Beijing 100871, China. [4]College of Mathematics and Physics, Beijing University of Chemical Technology, Beijing 100029, China. [5]Department of Chemical Engineering, University of Cambridge, Cambridge CB2 3RA, UK. [6]Energy Research Institute @ Nanyang Technological University, 50 Nanyang Avenue, Singapore 639798, Singapore. [7]These authors contributed equally: Xiao Ren, Tianze Wu. ✉e-mail: xuzc@ntu.edu.sg

ferromagnetic material from a multi-domain one towards a single-domain one[14]. Each magnetic domain is a small magnet, and theoretically, the spin-polarization promoted OER already occurs on these magnetic domains. Therefore, the enhancement is already achieved, and magnetization under an external magnetic field is not necessary for principle. A puzzle has been thus created, i.e., why magnetization under an external magnetic field can promote the OER, and what is the origin of the observed increment?

Here, we demonstrate that the origin of the OER enhancement under the magnetic field (magnetization) is from the disappeared magnetic domain walls. Using a series of NiFe thin film electrodes with controlled domain wall size, we found a strong correlation between the area of domain walls occupied on the thin films and their OER increment under magnetization. NiFe thin films were prepared by the magnetron sputtering method. The film thickness is varied from 200 to 800 nm, in which the magnetic domains are in a stripe shape, and the neighboring domains adopt antiparallel magnetization. The spin configuration within each single strip domain is identical, and the OER on every single domain follows the spin-facilitated pathways as reported earlier[10,14,15]. With such strip-type domains of NiFe films, the domain walls can be clearly identified, and the area on the film surface occupied by the domain wall can be quantified. By reducing the thickness of the NiFe film, the magnetic domains shrink in width, and accordingly, the area occupied by the domain wall increases. Under a sufficiently strong magnetic field, e.g., a saturated field, the domain wall disappears, and the NiFe thin film electrodes evolve into a single domain state. A strong correlation has been found between the degree of OER enhancement under magnetization and the area of domain walls originally occupied on the thin film surface.

## Results
### Prepared and characterization of NiFe thin films
The NiFe thin films were prepared by radio frequency (RF) magnetron sputtering on Si substrates. The cross-section of NiFe thin films was observed under the scanning electron microscope (SEM). As shown in Fig. 1a, NiFe thin films with thicknesses of 200, 300, 400, 500, 600, 700, and 800 nm, which are denoted as NiFe-200, NiFe-300, NiFe-400, NiFe-500, NiFe-600, NiFe-700, and NiFe-800, respectively. The energy-dispersive X-ray spectroscopy (EDX) characterization (Fig. 1b and Supplementary Fig. 2) shows the homogeneous dispersion of Ni and Fe in the as-prepared films. The Ni/Fe ratio in all as-prepared films keeps at about 4:1 (Supplementary Fig. 3 and Supplementary Table. 1). In the X-ray diffraction (XRD) patterns of these NiFe thin films (Fig. 1c), the diffraction peaks can be indexed for crystal planes (111), (200), and (220), which matches well with the fcc cubic structure of NiFe alloy (PDF#47-1405). A strong peak at around 59° is from the Si substrate. All these NiFe films are the same in crystal structure. The surface morphology of NiFe thin films was investigated by atomic force microscopy (AFM) under taping mode (Fig. 1d). The surfaces of all films show limited morphological differences and their roughness parameters are summarized in Supplementary Table 2.

### Magnetic force microscope (MFM) and micromagnetic simulation
The NiFe thin films were investigated under magnetic force microscopy (MFM). The magnetic domains in strip shape were observed, and the neighboring domains are in the opposite directions of magnetization (blue and red) with a domain wall (white) in between (Fig. 2a). Besides, clearly under MFM, the strip domains were found enlarging in width along with increasing film thickness. The domain in NiFe-800 is around 2.5 times wider than the domain in NiFe-200 (Fig. 2b). The strip domains in NiFe films were studied by micromagnetic simulations (see Supplementary Methods for details). Figure 2c and Supplementary Fig. 6 show the magnetization distribution at the equilibrium state over the NiFe models with different film thicknesses. The regions with

ordered magnetization are the strip domains. We further calculate the MFM images of those models using micromagnetic simulations, as shown in Supplementary Fig. 7. The simulated MFM images are consistent with the observation under MFM; that is, thickening the films is found to broaden the strip domains. As the size of strip domains in NiFe film increases, less area is occupied by domain walls (as shown with disordered magnetizations between neighboring domains). The computational results give the ratio of disordered magnetizations in different degrees ($|M_z|$) to all magnetizations in NiFe films, which is plotted as a function of the film thickness (Fig. 2d). The domain wall area is also analyzed from MFM data, which is summarized in Supplementary Table 2 and plotted against the film thickness in Supplementary Fig. 8. The results from micromagnetic simulations and MFM are consistent, which clearly indicates that the increase of the film thickness reduces the domain wall size of NiFe films and thus the area occupied by the domain wall decreases. These findings agree with earlier reports about thickness-dependent strip domains in NiFe films[16]. In the hysteresis loop of NiFe films (Fig. 2e), a typical linear slope during the scan from saturation field ($H_s$) to zero fields can be found in all loops[17]. Such slope persists because the anisotropy field is introduced during the relaxation of strip domains[18]. In addition, we extract the remanence ratio $M_r/M_s$ and saturation field $H_s$ from hysteresis loops, which are plotted versus the film thickness in Supplementary Fig. 9. The $M_r/M_s$ decreases while $H_s$ increases gradually as the film thickness increases, which strongly indicates enhanced strip domain structures[19]. The results confirm the strip domains throughout the NiFe films and are consistent with what has been observed by MFM.

### OER enhancement under magnetic field
As illustrated in Fig. 3a, the electrochemical measurements were carried out in an H-type cell with a three-electrode configuration, and a constant magnetic field was applied to the chamber with the working electrode (NiFe films). Cyclic voltammetry (CVs) were recorded for NiFe films with and without a magnetic field in $O_2$-saturated 1.0 M KOH. The field strength is held at 2000 Oe in the plane, which is strong enough for a complete in-plane magnetization of the NiFe films. All NiFe films were pretreated with 5 CV cycles to reach steady CV profiles before applying the magnetic field (Supplementary Fig. 10). It should be noticed that, in principle, a very thin surface layer of NiFe oxyhydroxide should be created upon the CVs pretreatment (Supplementary Figs. 11–14), which follows the substrate domain structure and can response to the external magnetic field through the pinning effect as reported early[6]. Figure 3b shows the steady CVs of NiFe films before and after applying the magnetic field. All potentials have been converted to the RHE scale with iR correction. As seen in the results, while applying the magnetic field, enhanced OER activities are found for all NiFe films. Notably, the degree of OER enhancement highly depends on the film thickness. As shown in the summary of the over-potential drop at 10 mA cm$^{-2}$ with and without the magnetic field (Fig. 3d), the NiFe-200 exhibits the biggest enhancement under a magnetic field, while the NiFe-800 shows the smallest one.

## Discussion
It has been recently recognized that the OER activity delivered by magnetic catalysts can be promoted under a magnetic field[4, 5,20,21]. When the magnetization degree of a magnetic catalyst becomes higher, larger OER enhancement can be observed[5,20]. Therefore, the magnetism of catalysts was believed to be responsible for OER enhancement under a magnetic field. However, this magnetization-caused improvement in OER is still under debate. This is because, for some cases, nearly no OER enhancement can be found under a magnetic field, even with a considerable magnetization degree[14]. The answer to this question is related to understanding the origin of the increment under the magnetic field. Here, we found that the OER enhancement under a magnetic field can vary greatly for NiFe films

 

with different thicknesses. For NiFe-800, the OER enhancement by magnetization is even negligible. All NiFe films in different thicknesses keep nearly the same crystal structure but with different magnetic domain structures. The domain wall proportion decreases as the NiFe film thickness increases. In Fig. 4a, we compare the area ratio occupied by the domain wall and the degree of OER improvement under magnetic field (in overpotential drop at 10 mA $cm_{film}^{-2}$) of NiFe films with different thicknesses, in which a clear correlation is found. The magnetic domains are the basic unit in ferromagnetic materials that respond to the external magnetic field (a strong enough field). In NiFe films, without an applied magnetic field, the neighboring strip domains show magnetization in opposite directions, which gives a zero net magnetization. As illustrated in Fig. 4b, the unpaired electrons in the domain wall are in a transition region in which the spin direction is affected by two neighbor domains with opposite magnetization

directions and thus are highly disordered[22,23]. The domain wall region becomes less in the overall thin film as the film becomes thicker, and the magnetic domain becomes bigger (Fig. 4c). By being magnetized under a strong enough magnetic field, all magnetic domains in NiFe films are aligned to the same direction as that of the external field, for which the films evolve from multi-domain ones to single-domain ones (Fig. 4c) and the domain walls involving disordered spins disappear.

In early theories about spin-related OER kinetics, it was proposed that ferromagnetic spin ordering on atoms in catalysts can optimize the exchange interaction of parallel spins to facilitate spin-polarized electron transfer and reduce the kinetic barrier in OER[8, 15,24]. These theories are established at the atomic scale and, in fact, tell the spin-polarization promotion mechanism on an individual magnetic domain. For a ferromagnetic material, the spin of unpaired electrons in the individual magnetic domains is completely aligned in one direction.

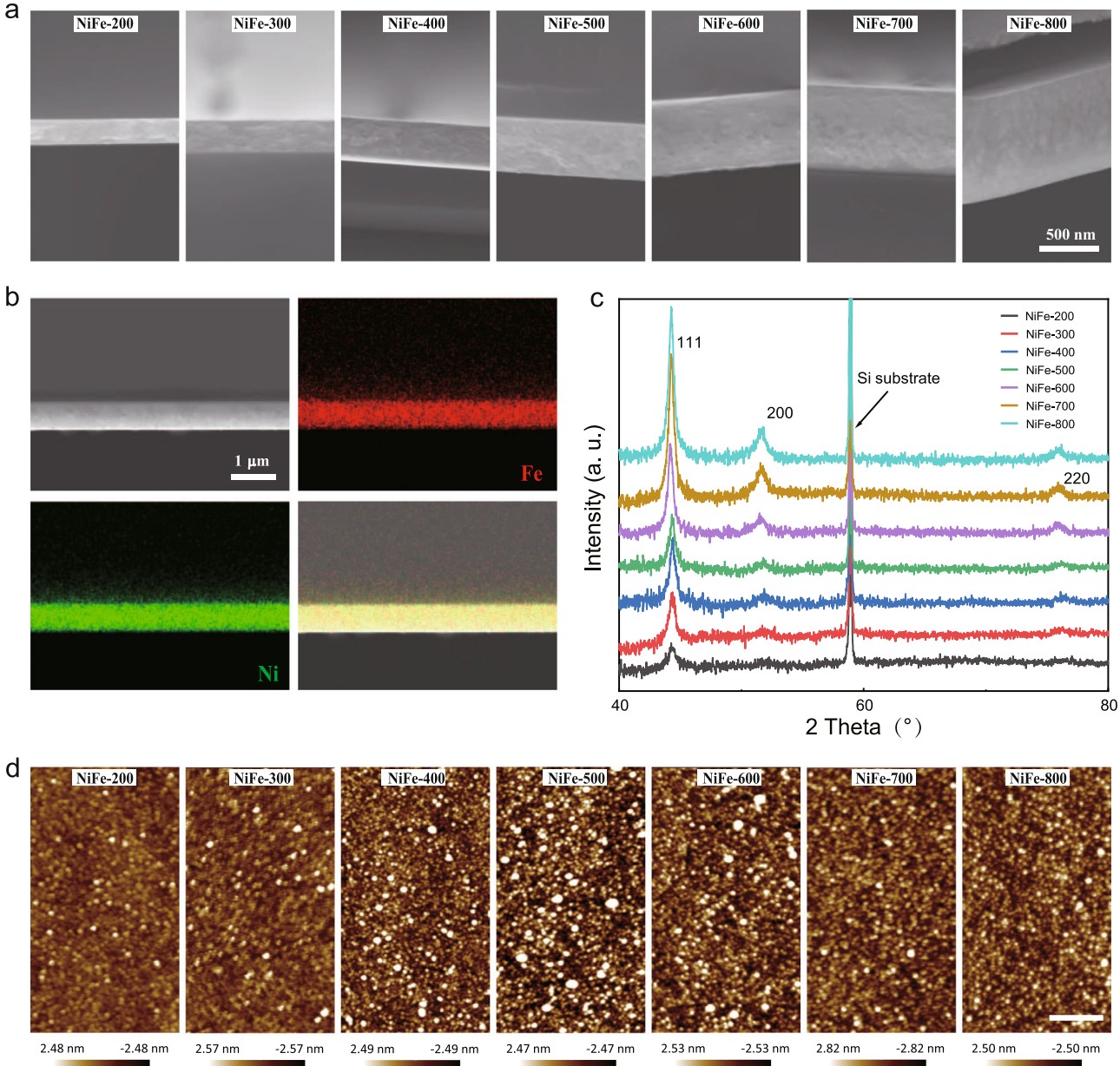

**Fig. 1 | Characterizations of NiFe thin films. a** The SEM images of the Cross section of NiFe thin films on Si substrates. The scale bar of those images is 500 nm. **b** The EDX mapping photos of NiFe-600 (the mapping photos of other films can be found in Supplementary Fig. 2). **c** The XRD patterns of the NiFe thin films. **d** Surface morphology of NiFe thin films obtained by AFM under tapping mode. The scale bars of those images are 400 nm.

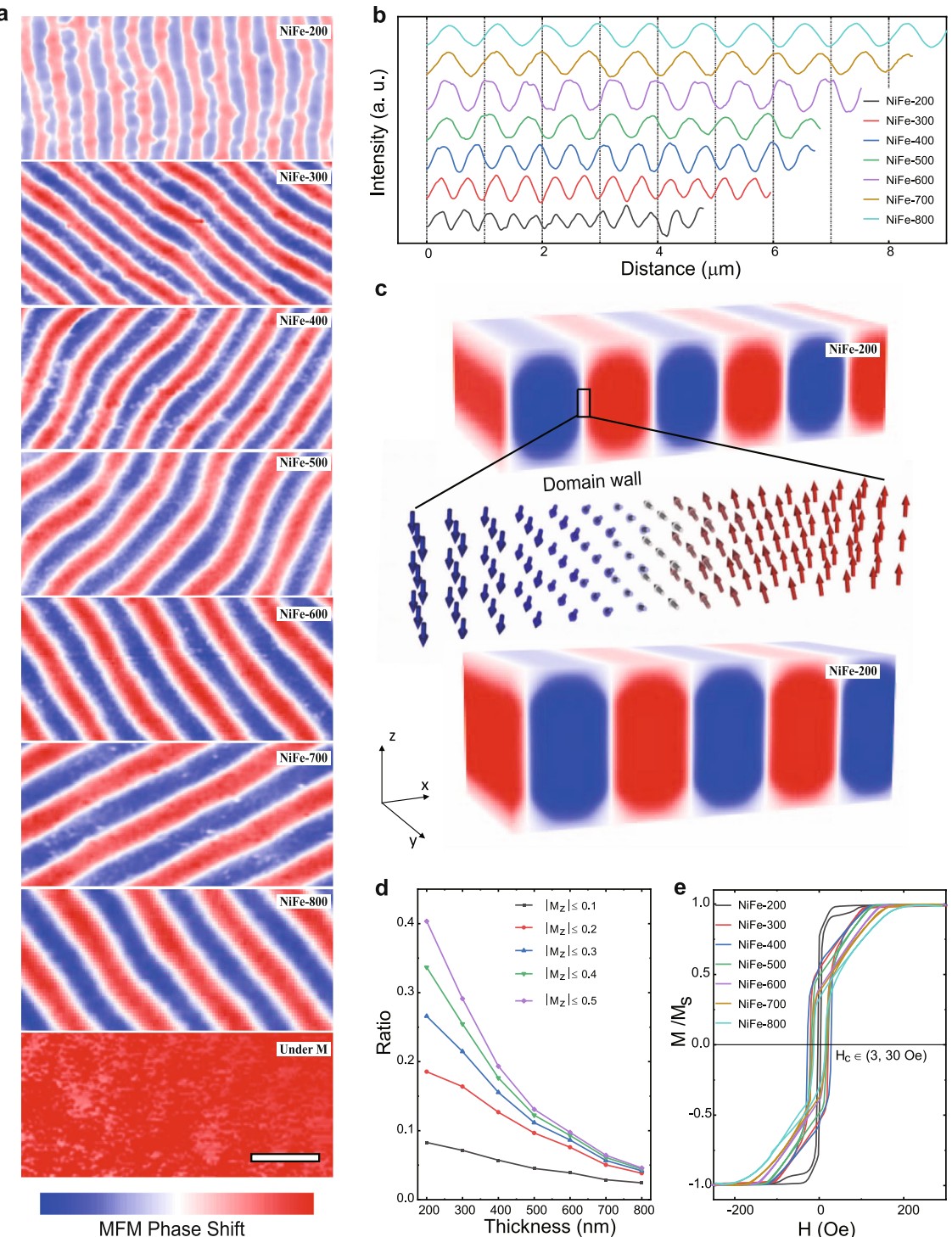

**Fig. 2 | Magnetic property characterizations of NiFe thin films. a** Zero-field MFM images of the magnetic domain structure of NiFe films with thicknesses of 200, 300, 400, 500, 600, 700, and 800 nm and a representative one (NiFe-600) for the magnetized films under an in-plane magnetic field of 2000 Oe. The scale bar of those images is 1 μm. **b** The phase line profiles acquired from MFM images of NiFe films. **c** Distributions of the $M_Z$ magnetization component over stripe domains of NiFe-200 and NiFe-300 by micromagnetic simulations. **d** The proportions of domain wall in NiFe films as summarized from micromagnetic simulation results. The $M_Z$ means the degree of deviation of magnetic distance along *the* z-axis direction. **e** Experimental in-plane hysteresis loops for different NiFe film thicknesses.

Therefore, the fact is that the spin-related promotion effect on OER has been achieved on each individual magnetic domain surface[10,11]. By applying a magnetic field, it is highly impossible to further increase the spin polarization within the individual magnetic domain[25]. The field should have a negligible effect on the net spin angular momentum on every unpaired electron, including those on the active sites, but can

only align all magnetic domains into an identical direction and eliminate the domain walls between domains. It further indicates that the OER enhancement by the magnetization is not due to the spin flipping of unpaired electrons on atoms, which is impossible. Instead, the fundamental link should exist at a size level larger than atoms, i.e., the magnetic domains and the domain walls in between. The correlation

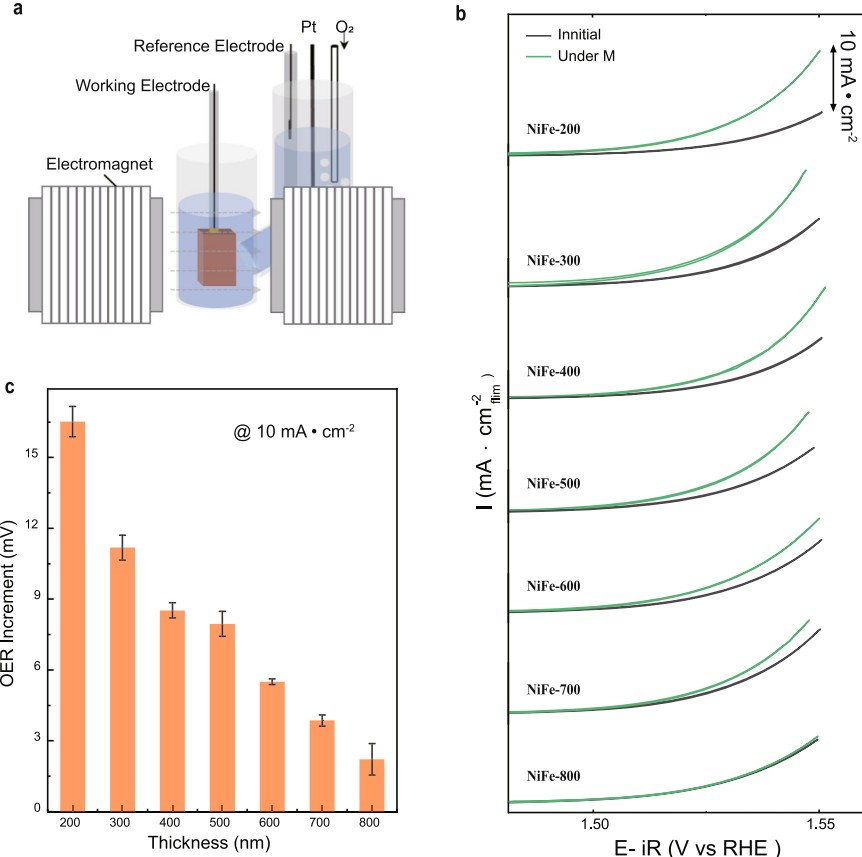

**Fig. 3 | Magnetization enhanced OER. a** The schematic illustration of the experimental set-up for electrochemical measurements under a constant magnetic field. **b** Cyclic voltammetry (CV) of NiFe thin films with different thicknesses at a scan rate of 10 mV s$^{-1}$ in O$_2$-saturated 1.0 M KOH with and without a constant in-plane magnetic field (2000 Oe). **c** The OER overpotential drop of NiFe films with the applied magnetic field. Error bars represent the standard deviation obtained from three independent measurements.

between the OER enhancement and the domain wall proportions indicates that such OER increment under magnetization is led by the evolution from a multi-domain to a single-domain state, in which the domain walls disappear, and the region occupied by the domain wall is reformatted into a region occupied by the magnetic domain. After reformatted, the region originally occupied by the domain wall changes to the magnetic domain region, in which the spin-related promotion effect can be realized. The proportion of domain walls in catalysts determines how much space it is for OER to be enhanced by magnetization. For example, the domain wall occupies a limited region in NiFe-800 film, and thus there is no notable OER improvement that can be observed by magnetization. With this understanding, it is reasonable that not all types of magnetic catalysts can give OER enhancement by magnetization. For NiFe-200 film, with the highest proportion of domain wall among all films, it gives the highest-degree OER increment under magnetization. We further establish the relationship between the relative magnetic domain wall area of those films and their electrochemical activity shown in Supplementary Fig. 27 (It should be noted that such correlation may not be extrapolated beyond the shown regime). The OER overpotential drop of NiFe films after magnetization is monotonically increasing with the relative area of the domain wall, which indicates the increment of OER activities of these NiFe films under magnetization originates from the disappeared magnetic domain walls. The findings in this study indicate that the ferromagnetic electrocatalysts with more domain walls are more likely to show OER enhancement by magnetization.

Overall, we have prepared NiFe films with thicknesses from 200 to 800 nm with controlled domain size and domain wall occupations. On the basis of such a well-defined domain structure, the domain wall area

has been quantified for each NiFe film. The area occupied by domain walls decreases as the film thickness increase. The domain wall disappears when completely magnetized under an external magnetic field of >2000 Oe, which leads to the evolution of NiFe films from multi-domain to single-domain ones. The region occupied by the domain wall is reformatted into a magnetic domain region, in which the spin-related promotion effect for OER can be realized. The increment of OER activities of these NiFe films under magnetization is determined by the proportions of surface regions occupied by domain walls in NiFe films before being applied with an external magnetic field. This study has addressed the origin of the OER increment on ferromagnetic catalysts. Our results show that the magnetic enhancement originates from domain wall dynamics caused by magnetization. To further understand the OER over domain wall, new computational approaches are needed to establish models in domain wall structures with consideration of inter-domain interactions at the atomic scale.

## Methods

### Films preparation

NiFe thin films in this work were prepared by radio frequency (RF) magnetron sputtering on naturally oxidized Si (1 1 1) substrates at a common base pressure of <5 × 10$^{-5}$ Pa, a work processing Ar pressure of 0.2 Pa, a power of 50 W, and a target substrate distance of 9 cm. The samples were deposited from a permalloy target with a nominal atomic composition of 20/80. During sputtering, at room temperature, the columnar crystals formed by vertical sputtering introduce shape anisotropy, which is perpendicular magnetic anisotropy[26,27]. With these parameters, we obtained a sputtering rate of 0.15 nm/s. This rate was calculated from a control sample with a mask in which the

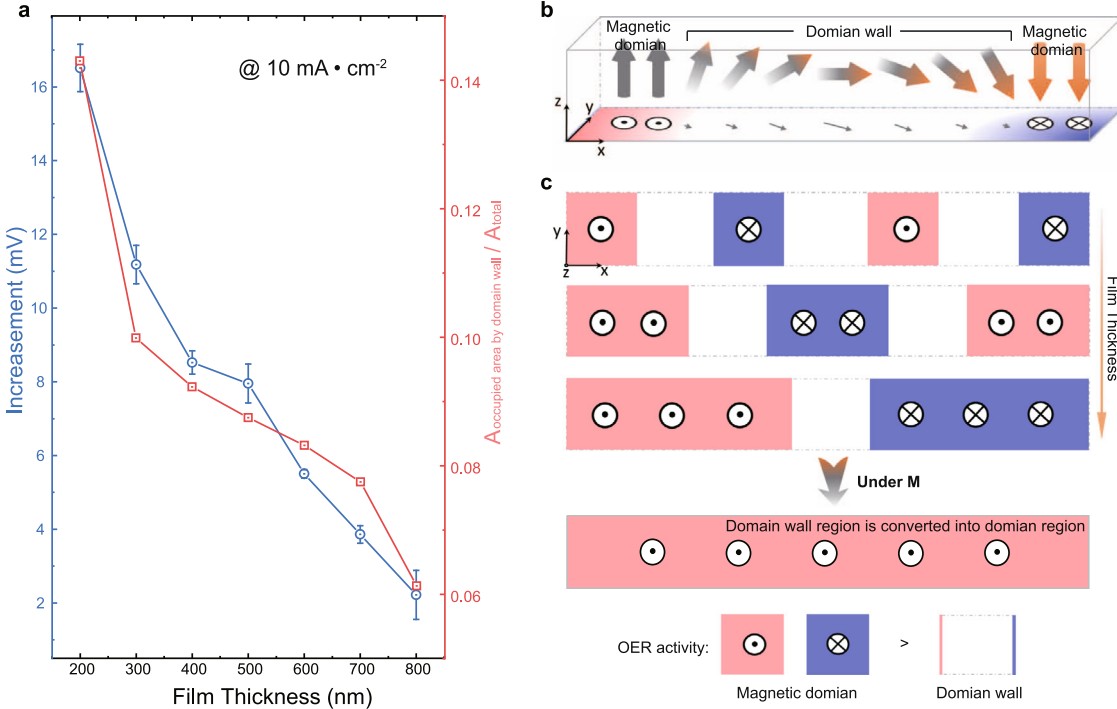

**Fig. 4 | The origin of magnetization-caused increment in OER. a** The OER overpotential drop of NiFe films after magnetization (left axis, blue) and the ratio of the surface area occupied by domain walls to the total surface area of NiFe films (right axis, red), as a function of film thickness. Error bars represent the standard deviation obtained from three independent measurements. **b** The profile of a domain wall in the NiFe film with arrows denoting the magnetization distribution.

**c** The schematics of magnetic domains and domain walls in NiFe films with different film thicknesses. Applied with a strong enough magnetic field, all domain walls in NiFe films disappear, for which the films evolve from multi-domain ones to single-domain ones, and the surface is reformatted from one with single-domains and domain walls into one with only single-domain occupied.

height step was measured using an atomic force microscope. Seven films were prepared with different thicknesses: 200, 300, 400, 500, 600, 700, and 800 nm.

## Electrochemical characterization

The OER tests were operated in a three-electrode cell with a NiFe film as the working electrode (WE), a platinum foil counter electrode, and a Hg/HgO (1.0 M KOH) reference electrode. The schematic diagram of working electrode preparation shows in Supplementary Fig. 1. Au was deposited on the NiFe films where contact with the Pt electrode holder. Then, the film as a working electrode was attached to a Pt electrode holder. Subsequently, the back and sides of the electrode were covered with a non-conductive, chemically resistant epoxy (Omegabond 101). An effective electrode surface (0.2 cm × 0.2 cm; area: 0.04 cm$^2$) was exposed to 1.0 M KOH during electrochemical measurements. The electrochemical measurements were carried out by a Biologic SP-150 potentiostat. The KOH electrolyte with a concentration of 1.0 M was prepared using deionized water (>18 MΩ cm) and KOH pellets (99.99% purity, Sigma-Aldrich). Oxygen was bubbled to ensure $O_2$/$OH^-$ equilibrium at 1.23 V vs. RHE. All potentials were converted to RHE scale with iR correction according to the equation: $E$ (vs. RHE) = $E$ (vs. Hg/HgO) + 0.059 × pH + 0.098·$iR$, where $i$ is the current and $R$ is the electrolyte resistance, determined by high-frequency AC impedance (-15 Ω for 1.0 M KOH). All electrochemical data were normalized using film surface area (Supplementary Table 2).

## Materials characterizations

The X-ray diffraction (XRD) patterns of these films were recorded by a Bruker D8 diffractometer at a scanning rate of 2° min$^{-1}$, under Cu-K$_\alpha$ radiation ($\lambda$ = 1.5418 Å). Scanning electron microscopy (SEM) and energy-dispersive X-ray spectroscopy (EDX) were performed using a HITACHI S5000 with an energy-dispersive analysis system, Bruker

XFlash 6|60. DC magnetization measurements were performed on a Superconducting Quantum Design (SQUID) magnetometer (MPMS-XL). The SQUID measurements of the magnetization of samples as a function of the magnetic field were carried out at 300 K in fields between −2000 Oe and +2000 Oe. Film surface morphologies were examined by atomic force microscopy (AFM) under tapping mode (Asylum Research). The Cypher AFM/MFM manufactured by Asylum Research was performed for MFM experiments. The MFM tips used were PPP-MFMR from NANOSENSORS with 300 Oe coercivity, 300 emμ cm$^{-3}$ magnetic moment, and magnetic resolution better than 50 nm, optimized for non-perturbative magnetic imaging with high spatial resolution.

## Micromagnetic simulations

The domain structures formed in thin films were investigated by micromagnetic simulation of a thin film model with different thicknesses, in which perpendicular magnetic anisotropy with slight tilt was applied. Micromagnetic simulations were performed using the mumax3 software package[28]. The micromagnetic simulations are based on the Landau-Lifshitz-Gilbert (LLG) equation with the magnetic Hamiltonian described as

$$\mathcal{H} = -A \sum_{<i,j>} \mathrm{m}_i \cdot \mathrm{m}_j - K_{in} \sum_i \left(\mathrm{m}_i\right)^2 - K_p \sum_i \left(\mathrm{m}_i\right)^2 - \sum_i B \cdot \mathrm{m}_i \quad (1)$$

where $\mathrm{m}_i$ represents the normalized spin at the site $i$, $|\mathrm{m}_i| = 1$. $A$, $K_{in}$, $K_p$ represent the amplitude of exchange interaction, in-plane and perpendicular magnetic anisotropy. $B$ is the external magnetic field.

It should be noted that Mumax3 does not provide two different kinds of uniaxial anisotropy straightforwardly. Fortunately, the parameter $K_u$ is a vector so that we can consider in-plane and perpendicular

magnetic anisotropy are two components of $K_u$. The form of $K_u$ seems like $K_u = (K_{in}, 0, K_p)$.

We consider the model of a thin film with magnetic parameters corresponding to that of the typical permalloy $Ni_{80}Fe_{20}$ film[29]—the saturation magnetization $M_s = 1000$ emu/cm$^3$, the exchange constant $A = 1 \times 10^{-6}$ erg/cm, and the in-plane uniaxial anisotropy $K_{in} = 3.25 \times 10^3$ erg/cm$^3$. The formation of stripe domains in NiFe films, in general, is caused by competing exchange interaction energy, perpendicular magnetic anisotropy energy, and demagnetization energy. When the thickness varies, the perpendicular magnetic anisotropy energy and demagnetization energy may both vary dependently. The demagnetization energy can be calculated automatically during the simulation. The perpendicular magnetic anisotropy energy should be set manually. Here we make an estimation to take the thickness-dependent perpendicular anisotropy into consideration: the perpendicular anisotropy of NiFe films increases linearly from $2.5 \times 10^5$ erg/cm$^3$ to $4 \times 10^5$ erg/cm$^3$ in the range of 200–800 nm[19]. The in-plane anisotropy was oriented along the $x$-axis, while the perpendicular anisotropy lay along the $z$-axis (normal to the film plane). The size of each discretization cell was $4 \times 4 \times 4$ nm. We set a $2048 \times 50 \times N$ mesh with in-plane periodic boundary conditions to avoid the boundary effect. $N = 50, 75, 100, 125, 150, 175, 200$ correspond with the film thickness 200–800 nm. Note that, in our simulations, the length of the films along the $x$-axis must be much longer than the width of domains to reduce the impact of integer periods on magnetic domains. The approach can, consequently, accurately get the volume ratios of domain walls to the whole system. Starting with a random initial magnetic state, the system is applied with a small external magnetic field along *the $y$*-axis to speed up the evolution. Then the field is repealed to obtain the ground states. All the data were extracted from the ground states. Technical details on the image attached can be extracted from the mumax3 input file used to generate the data:

```
Ms: = 1000 // emu/cm3
Msat = Ms*1000 // A/m
A: = 1e−6 // erg/cm
Aex = A*1e−5 //J/m
Hin: = 6.5 // Oe
Kin: = Hin * Ms/2 * 1e−1// J/m3
Hp: = 517 // Oe
Kp: = Hp * Ms/2 * 1e−1 // J/m3
Ku1 = sqrt(Kin * Kin + Kp * Kp)
anisU = vector(Kin, 0, Kp)
alpha = 0.025
dx: = 4e-9
dy: = 4e-9
dz: = 4e-9
nx: = 2048
ny: = 100
nz: = 50
SetGridSize (nx, ny, nz)
SetCellSize (dx, dy, dz)
SetPBC (4, 4, 0)
m = randommag ()
B_ext = vector (0, 1, 0)
Relax ()
B_ext = vector (0, 0, 0)
Run (2e−8)
Relax ()
Save (m)
```

## Data availability

The data that support the findings of this study are available from the corresponding author upon reasonable request.

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

## Acknowledgements
The authors thank the financial support from the Singapore Ministry of Education Tier 2 Grant (MOE-T2EP10220-0001). Authors appreciate the Facility for Analysis, Characterization, Testing, and Simulation (FACTS) at Nanyang Technological University for materials characterizations. This work was partially supported by the National Research Foundation (NRF), Prime Minister's Office, Singapore, under its Campus for Research Excellence and Technological Enterprise (CREATE) program.

## Author contributions
Z.X. conceived the hypothesis and original idea. Z.X. and X.R. designed the experiments. and L. P. prepared the materials, and X.R. performed most characterizations. J. M. and X.R. performed the AFM/MFM measurement. Z.G. contributed to the micromagnetic simulations. R.X. plotted and prepared all figures. R.X., T.W., and Z.X. wrote the manuscript. H.Y. and H.G. provided characterization instruments. F.B.D. and A.F. assisted CV analysis.

## Competing interests
The authors declare no competing interests.
