## [Peer review file · Nature Communications]

REVIEWER COMMENTS

Reviewer #1 (Remarks to the Author):

Thanks to the authors for the hard work and discussions on their manuscript. From my point of view and after reading all the responses to the reviewers, I think that the manuscript should be accepted at n. communications

But, before acceptance, I'd like some clarification on the electrochemical setup (I apologise, I missed it in the previous review).

In the schematic setup, videos and photos, I noted that the authors used the reference electrode on the cathode side. So I would like to know why the authors used this configuration. Normally, in electro-catalysis the reference electrode (RE) must be close to the working electrode (WE) (in this case NiFe alloy) to know the voltage changes during the reaction (in this case OER). So, I'm surprised or maybe I don't understand the motivation to track the changes in the WE, just by placing the RE near the cathode side.

Reviewer #2 (Remarks to the Author):

In the revised manuscript, Ren et al. have addressed the points of discussion raised by the referees. I recommend publication after addressing the few remaining points below.

- In response figure R15, the authors infer Ni³⁺ states, i.e. NiOOH. Yet based on a Pourbaix diagram, one would expect this phase to be present only at high oxidizing potentials, e.g. during OER. But once the potential is removed, one would expect a Ni(OH)₂, and therefore Ni²⁺ oxidation state. This should be further elaborated.

- The authors now included changes in EIS-derived resistances. It would help if they could also demonstrate that magnetoresistance is not the cause of the enhancement by showing resistivity vs. magnetic field measurements.

- In the revised version, there are grammatical mistakes in the added sentences.

Reviewer #4 (Remarks to the Author):

I thank the authors for their efforts to answer the concerns raised in the previous review round. However, after careful consideration, I am still not convinced that the conclusions are fully supported by the data.

The oxygen evolution reaction happens at the surface of the catalyst, and hence any increase in the efficiency must be explained by surface properties. As I will show below, this is currently not the case. MFM is a technique which measures magnetic stray fields, which for the structures considered here are generated by a small out-of-plane tilt of the magnetization at the surface, originating in the bulk magnetization structure. The bulk magnetization consists of large out-of-plane domains, separated by sharp domain walls. This structure is slightly reflected in the magnetization at the surface, which has a small out-of-plane tilt of about 10%. This means that the MFM images are blind to about 90% of the magnetization texture at the surface. The authors compare the MFM images to the amount of out-of-plane magnetization extracted from micromagnetic simulations (and have clarified in their response that they calculate this also over the entire bulk of the simulations, instead of only at the surface). This means that both quantities are seemingly in agreement, but give little information on what is happening at the surface.

As I have already mentioned in my previous review report, the surface contains in-plane closure domains with a more complex domain wall structure than the one visible in the MFM pictures. To illustrate this point, I have attached four figures of the same structure, two with a color scheme similar to the one used by the authors which only shows the out-of-plane magnetization in red/blue and the in-plane in white (which therefore hides all in-plane magnetization structures); and two with a color scheme which shows the out-of-plane components in white/black and the different in-plane magnetization directions in different colors. As is clearly visible, the surface is magnetized almost fully in-plane with a much less clear-cut division in domains and domain walls, but rather display a continuously changing magnetization direction.

Furthermore, I have repeated the authors' simulations and quantified the out-of-plane magnetization **only** at the surface layer, instead of in the bulk, and have found that this varies much less strongly (from about 10% for 200 nm thick films to about 9% for 800 nm thick films).

The authors do find an interesting correlation between the stray fields at the surface (reflecting the underlying domain structure in the bulk of the material) and the OER enhancement, but because at the surface there is a very different domain structure present than the one with which the authors correlate their findings, the current explanation based on the relative area occupied by domains and domain walls cannot be the complete picture.

Moreover, the correlation, as depicted in figure 4, is visually misleading, because both axes have an offset with respect to the origin, and when extrapolating the data towards thicker films, where the domain wall area would fall below 5%, a negative OER increase would even be expected, which clearly would contradict the presented explanation.

Finally, besides the observation that there might be a correlation (and please note that correlation does not imply causation) between the OER increase and the stray fields, there is no explanation of the microscopic process underlying this enhancement (which was the main claim of the paper). The conclusion that "This study has addressed the origin of the OER increment on ferromagnetic catalyst" is therefore unsubstantiated.

The only thing that comes close to an explanation, i.e. lines 189 and following: "... the unpaired electrons in domain wall are in a transition region in which the spin direction is affected by two neighbor domains with opposite magnetization direction and are highly disordered." is unconvincing. Reference 22 which was cited in this context also does not contain any statements on disorder in the spin directions. The OER is a reaction that takes place on atomic length scales, and therefore should be influenced by the local environment only. In contrast, the domain walls at the surface vary much more gradually over length scales of about 20 nm, and except for not being exactly aligned with their neighbors, the spins are not at all disordered.

A few technical comments:

- The description of the anisotropy used in the simulations is not correct. There is a difference between a system with 2 uniaxial anisotropies, i.e. a biaxial anisotropy with 4 easy directions of the magnetization, and the system described by the authors consisting of a single uniaxial anisotropy which is slightly tilted such that the anisotropy vector has both an in-plane and out-of-plane component. The top of the method section makes clear that the latter implementation was used, but the description in the remainder of the manuscript is ambiguous.

- Figure 2 in the manuscript still shows M_z in the legend, despite the correction to "MFM phase shift" that was mentioned by the authors on page 31 of their response.

In short, I still find the conclusions insufficiently supported by the results, and find a major revision necessary before the manuscript can be considered for publication, either in this journal or elsewhere.

In my opinion, publication in this journal requires a deeper understanding that goes beyond the description of an observed correlation.

Reviewer #1 (Remarks to the Author):

Thanks to the authors for the hard work and discussions on their manuscript. From my point of view and after reading all the responses to the reviewers, I think that the manuscript should be accepted at n. communications

But, before acceptance, I'd like some clarification on the electrochemical setup (I apologise, I missed it in the previous review).

In the schematic setup, videos and photos, I noted that the authors used the reference electrode on the cathode side. So I would like to know why the authors used this configuration. Normally, in electro-catalysis the reference electrode (RE) must be close to the working electrode (WE) (in this case NiFe alloy) to know the voltage changes during the reaction (in this case OER). So, I'm surprised or maybe I don't understand the motivation to track the changes in the WE, just by placing the RE near the cathode side.

Response: Thanks for the comment. We designed the cell in such a way to ensure that the two heads of the electromagnet are very close to each other. This will help to get the desired field strength at a small power input to the electromagnet. Otherwise, a high-power input has to be applied and it will result in a heating effect from the electromagnet to the electrochemical cell in between.

Reviewer #2 (Remarks to the Author):

In the revised manuscript, Ren et al. have addressed the points of discussion raised by the referees. I recommend publication after addressing the few remaining points below.

- In response figure R15, the authors infer Ni³⁺ states, i.e. NiOOH. Yet based on a Pourbaix diagram, one would expect this phase to be present only at high oxidizing potentials, e.g. during OER. But once the potential is removed, one would expect a Ni(OH)₂, and therefore Ni²⁺ oxidation state. This should be further elaborated.

Response: Thanks for the reviewer's suggestion. We have fit the 2p_{2/3} orbitals of Ni and Fe as shown in the figure below (Figure R1 in this response letter). The metallic Fe and Ni (labeled as Fe⁰ and Ni⁰) with narrow peaks can be found in the film samples. After the electrochemical tests, the peaks of Fe⁰ become weakened and the XPS spectra of Ni 2p_{3/2} can be fitted with two characteristic peaks at 855.2 and 856.2 eV, which are associated with the formation of Ni(OH)₂ and NiOOH, respectively (Surf. Interface Anal. 2009, 41, 324–332). The result indicates that the NiFeO_xH_y is formed on the surface. These data have been updated in SI (Line 174, page 15 in SI, marked in red)

Figure R1 & Supplementary Fig.12. The Ni and Fe 2p for all NiFe films before and after the electrochemical tests.

- The authors now included changes in EIS-derived resistances. It would help if they could also demonstrate that magnetoresistance is not the cause of the enhancement by showing resistivity vs. magnetic field measurements.

Response: Thanks for the suggestion. Figure R2 shows the magnetoresistance (MR) of three samples under the magnetic field. The MR of the NiFe films is very small. We did not find any correlation between the MR change and the film thickness. Indeed, the measurement demonstrates that MR is not the cause of the enhancement.

Figure R2. R-H curves for NiFe-200, NiFe-300 and NiFe-400. The magnetic field applied are in-plane.

- In the revised version, there are grammatical mistakes in the added sentences.

Response: We have checked writing in the revised manuscript and corrected mistakes (marked in red in the revised manuscript).

Reviewer #4 (Remarks to the Author):

I thank the authors for their efforts to answer the concerns raised in the previous review round. However, after careful consideration, I am still not convinced that the conclusions are fully supported by the data.

The oxygen evolution reaction happens at the surface of the catalyst, and hence any increase in the efficiency must be explained by surface properties. As I will show below, this is currently not the case. MFM is a technique which measures magnetic stray fields, which for the structures considered here are generated by a small out-of-plane tilt of the magnetization at the surface, originating in the bulk magnetization structure. The bulk magnetization consists of large out-of-plane domains, separated by sharp domain walls. This structure is slightly reflected in the magnetization at the surface, which has a small out-of-plane tilt of about 10%. This means that the MFM images are blind to about 90% of the magnetization texture at the surface. The authors compare the MFM images to the amount of out-of-plane magnetization extracted from micromagnetic simulations (and have clarified in their response that they calculate this also over the entire bulk of the simulations, instead of only at the surface). This means that both quantities are seemingly in agreement, but give little information on what is happening at the surface.

As I have already mentioned in my previous review report, the surface contains in-plane closure domains with a more complex domain wall structure than the one visible in the MFM pictures. To illustrate this point, I have attached four figures of the same structure, two with a color scheme similar to the one used by the authors which only shows the out-of-plane magnetization in red/blue and the in-plane in white (which therefore hides all in-plane magnetization structures);

and two with a color scheme which shows the out-of-plane components in white/black and the different in-plane magnetization directions in different colors. As is clearly visible, the surface is magnetized almost fully in-plane with a much less clear-cut division in domains and domain walls, but rather display a continuously changing magnetization direction.

Furthermore, I have repeated the authors' simulations and quantified the out-of-plane magnetization *only* at the surface layer, instead of in the bulk, and have found that this varies much less strongly (from about 10% for 200 nm thick films to about 9% for 800 nm thick films).

The authors do find an interesting correlation between the stray fields at the surface (reflecting the underlying domain structure in the bulk of the material) and the OER enhancement, but because at the surface there is a very different domain structure present than the one with which the authors correlate their findings, the current explanation based on the relative area occupied by domains and domain walls cannot be the complete picture.

Response: Thanks for the further questions.

First, we agree that the oxygen evolution reaction happens at the surface of the catalyst. But, the well-accepted thicknesses of “surface” has a few nanometers to tens of nanometers by previously reported (Nature Chem 9, 457–465 (2017); Sci. Adv. 3, e1603206 (2017)). In this work, the minimum depth of the catalyst film surface involved in the reaction is about 11 nm as shown in Supplementary 14. In micromagnetic simulations, the depth of the surface is determined by the set cell size. We set the cell in this work based on the reviewer's suggestion in which the depth of this surface is about 4 nm. This thickness in micromagnetic simulations are much less than the thickness of the actual reaction surface region.

Second, we appreciate the simulated domain structure done by the reviewer and fully agree. The arrangement of magnetic moments on the surface is more complex than in the bulk, but these two are directly correlated, that is, the periodic arrangement of the magnetic domains and domain walls directly determines the magnetic configuration of the surface. The in-plane variation of the surface magnetic moment over a full stripe period can be determined as $-\pi/2, 0, \pi/2, 0, -\pi/2$. The in-plane variation period of the surface magnetic moment remains constant as the domain width changes as depicted in **Figure R3**. The average angle of the magnetic moment within a period is related to the stripe period. The direction of the magnetic moment inside the bulk domain changes slowly, the direction of the magnetic moment at the domain wall changes faster. The magnetic moment at the surface follows the same rule. That is, the direction of the magnetic moment at the surface locate around $|\pi/2|$ changes faster, and locate around 0 changes slowly.

Figure R3. Equilibrium magnetization distribution in bulk and surface obtained from micromagnetic simulation of NiFe-200 film.

Third, it should be emphasized that the OER enhancement is from the magnetization. The reviewer paid much attention on the status without magnetization but should not ignore the change after magnetization, which should be paid more attention on. This is a clear case of the amount of change in the magnetic state after the magnetization (application of the magnetic field). We further calculated the rate of change of the magnetic moment along the x-direction of the surface shown in **Figure R4**. The variation of magnetic moment is small in the domain region and large in the domain wall region. The rate of change of the magnetic moment at the surface corresponding to the domain wall shows a clear thickness dependence, i.e., the thicker the film and the wider the stripe, the smaller the rate of change of the magnetic moment. When the magnetic field is applied, all magnetic moment is pulled parallel. The thinner the sample, the greater the change in magnetic moment. It is exactly what have been observed.

Figure R4. a. The difference between two adjacent magnetic moments at different positions in the X-direction on the surfaces of samples with different thickness; **b.** The average rate of change of magnetic moment over a full stripe period.

Overall, the thicker the film the higher the polarization (the smaller domain walls occupied), the smaller the OER enhancement under magnetic field.

Moreover, the correlation, as depicted in figure 4, is visually misleading, because both axes have an offset with respect to the origin, and when extrapolating the data towards thicker films, where the domain wall area would fall below 5%, a negative OER increase would even be expected, which clearly would contradict the presented explanation.

Response: Typically, the magnetic moments of NiFe sample mainly lie in the plane of the thin film due to the relatively strong demagnetization energy of film. However, an out-of-plane magnetization component of NiFe film arises above a critical film thickness, and an array of oscillating 'up and down' magnetic stripe domains forms due to the predominant factor in the perpendicular anisotropy. We have selected samples with significant variation in stripe domain width from 200-800 nm of thickness. Extending this correlation to thickness beyond the designated range will be problematic in the model films, because in thinner or thicker samples there may be formation of domains in non-stripe structure.

Finally, besides the observation that there might be a correlation (and please note that correlation does not imply causation) between the OER increase and the stray fields, there is no explanation of the microscopic process underlying this enhancement (which was the main claim of the paper). The conclusion that "This study has addressed the origin of the OER increment on ferromagnetic catalyst" is therefore unsubstantiated.

The only thing that comes close to an explanation, i.e. lines 189 and following: "... the unpaired electrons in domain wall are in a transition region in which the spin direction is affected by two neighbor domains with opposite magnetization direction and are highly disordered." is unconvincing. Reference 22 which was cited in this context also does not contain any statements on disorder in the spin directions. The OER is a reaction that takes place on atomic length scales, and therefore should be influenced by the local environment only. In contrast, the domain walls at the surface vary much more gradually over length scales of about 20 nm, and except for not being exactly aligned with their neighbors, the spins are not at all disordered.

Response: We have to strongly disagree with the reviewer. The microscopic process of spin-polarized OER has been reported in previous works (Phys. Chem. Chem. Phys., 2017, 19, 20451; Journal of Catalysis 2018, 361, 331–338; Phys. Chem. Chem. Phys., 2019, 21, 2977-2983; J. Phys. Chem. C 2019, 123, 9967-9972; Nat Communications, 2021, 12, 2608; ACS Catal. 2021, 11, 14249–14261; Current Opinion in Electrochemistry 2021, 30, 100804, etc.). Based on those understandings, one cannot explain why magnetization can enhance the activity. Because a ferromagnetic catalyst consists of many small domains, each of which can be treated as a small magnet. On these small magnets, the spin-polarized OER already exists as indicated in those studies. **Respectively saying, if the reviewer really understands the background of this direction, the reviewer should have the same question where the OER enhancement comes from under the magnetization.** This work demonstrates the origin of the enhancement. Thus, we strongly disagree with the reviewer about his/her opinion.

In addition, we refer to the Ref. 22 schematic diagram (the Figure 1 in reference) to illustrate the characteristics of domain and domain wall. Considering the suggestion of the reviewer, we added the reference(C. Kittel, Rev. Mod. Phys. 21, 541 (1949).) in the revised manuscript marked in red.

A few technical comments:

- The description of the anisotropy used in the simulations is not correct. There is a difference between a system with 2 uniaxial anisotropies, i.e. a biaxial anisotropy with 4 easy directions of the magnetization, and the system described by the authors consisting of a single uniaxial anisotropy which is slightly tilted such that the anisotropy vector has both an in-plane and out-of-plane component. The top of the method section makes clear that the latter implementation was used, but the description in the remainder of the manuscript is ambiguous.

Response: Thanks. We have corrected the description “in-plane and perpendicular magnetic anisotropy were applied” to “perpendicular magnetic anisotropy with slight tilt were applied” at the method section (marked in red in revised manuscript).

- Figure 2 in the manuscript still shows Mz in the legend, despite the correction to "MFM phase shift" that was mentioned by the authors on page 31 of their response.

Response: Thanks for pointing this out. We have updated the Figure 2 in the revised manuscript.

Line 106, page 6, marked in red:

Figure 2. Magnetic property characterizations of NiFe thin films. a, Zero-field MFM images of the magnetic domain structure of NiFe films with thickness of 200, 300, 400, 500, 600, 700, and 800 nm and a representative one (NiFe-600) for the magnetized films under magnetic field of 2000 Oe. The scale bar of those images is 1 μm . **b**, The phase line profiles acquired from MFM images of NiFe films. **c**, Distributions of the M_z magnetization component over stripe domains of NiFe-200 and NiFe-300 by micromagnetic simulations. **d**, The proportions of domain wall in NiFe films as summarized from micromagnetic simulation results. The M_z means the degree of deviation of magnetic distance along Z-axis direction. **e**, Experimental in-plane hysteresis loops for different NiFe film thicknesses.”

In short, I still find the conclusions insufficiently supported by the results, and find a major revision necessary before the manuscript can be considered for publication, either in this journal or elsewhere. In my opinion, publication in this journal requires a deeper understanding that goes beyond the description of an observed correlation.

REVIEWER COMMENTS

Reviewer #2 (Remarks to the Author):

In the revised manuscript, Ren et al. have addressed the points of discussion raised by the referees. The points I raise have been answered sufficiently.

I would also like to comment on the discussion between the authors and the referee 4:

1. Surface magnetization. I find this discussion very important. The authors observe that the surface is actually a NiOxHy layer (about 11 nm according to TEM). In their previous work (DOI 10.1038/s41467-021-23896-1), they observe a spin-pinning in such a (paramagnetic) surface layer, I included a snapshot from this publication below.

In the current manuscript, they discuss the “The proportion of domain wall occupied surface before magnetization” and its correlation with OER activity changes. Reviewer 4 pointed out that the surface of the Ni/Fe alloy will have a different magnetization from the bulk.

- a. I do not agree with the authors’ response that the surface layer with different magnetization is thinner than the “active surface layer” such that this surface-magnetization-effect should not play a role.
- b. I think the situation is more complex:

In this schematic, I took the “bulk” picture from the manuscript and added the two separate surface layers. The key question arising from the debate between the referee and the authors and from the authors’ previous work is: what will be the spin pinning in the (Ni/Fe)OxHy surface layer

2. Role of spin order: I agree with the authors that previous theoretical work (as listed in the response) provided a baseline for the argument that “material A with long-range FM order” has a higher OER activity than the same “material A without long-range FM order” due to changes in the reaction energetics and charge transfer kinetics which are influenced by the quantum spin exchange interactions (QSEI).
I find it very relevant to combine this nanoscopic picture with the mesoscopic picture including the domain walls. Considering the area of the surface that can be considered as “material A with long-range FM order” makes sense to me.
However, I agree with reviewer 4 that the surface magnetization deserves closer attention in the manuscript. I invite the authors to include such a consideration in the current manuscript, essentially combining their current work, the reviewer 4 input, and their previous work to arrive at a more complete picture.
3. I would be interested to learn what this means for the enhancement mechanism in the authors’ and the editors opinion. Is the OER, which is located at the surface, influenced by the magnetic field of the underlying domains? Or does the spin pinning on top of the domain

walls change when magnetizing the samples, leading to a change in magnetic order in the active surface layer?

I think these additions will make the paper much stronger and help resolve the debate between the authors and referee 4.

Reviewer #4 (Remarks to the Author):

I thank the authors for their detailed and respectful reply to my comments.

As stated in my first review, my background is in magnetism and I viewed the manuscript through this lens, which required additional clarifications to interpret the authors' results and claims correctly.

Given all information presented to me by the authors in their rebuttal, I can agree to the publication of the manuscript if the authors can clarify one final question and add a few minor things to the manuscript to improve its readability for the broad readership of the journal.

- Please add a small discussion on the fact that what is considered the "surface" for these reactions can be up to tens of nanometers deep into the material (as is indeed shown in the provided references) and place this in the context of the observed magnetization state (the MFM shows the magnetostatic fields "leaking through" from the bulk, whereas the surface (as defined as the monolayer thick boundary of the magnet), has a mostly in-plane arrangement that shows a continuous rotation). With this new information, it is now clear to me that my distinction between bulk and surface was not in line with how the surface should be considered for this reaction.

On a related, but important note, I agree that the presented data shows that the OER enhancement is related to the size of the domain walls in the samples. However, I still don't fully understand how these domain walls negatively affect the OER. The magnetization of the domain walls is no different than it is in the domains, it just lies in a different, gradually changing, direction. One explanation I can think of, given the clarification above, is that this reaction somehow probes a surface that is not only a few tens of nm deep, but also has such a large lateral dimension. On such length scales, the magnetization in the domain wall regions indeed cancels out (to a certain extent), which could explain a reduced local magnetization and hence the experimental data. Can the authors confirm that this is also their understanding (and if not, please provide an alternative explanation) and add this to the manuscript?

- Finally, please add a small note warning the readers that the results of figure 4 should not be extrapolated beyond the shown regime.

Response to Referees' Comments

Reviewer #2 (Remarks to the Author):

In the revised manuscript, Ren et al. have addressed the points of discussion raised by the referees. The points I raise have been answered sufficiently.

I would also like to comment on the discussion between the authors and the referee 4:

1. Surface magnetization. I find this discussion very important. The authors observe that the surface is actually a NiOxHy layer (about 11 nm according to TEM). In their previous work (DOI 10.1038/s41467-021-23896-1), they observe a spin-pinning in such a (paramagnetic) surface layer, I included a snapshot from this publication below.

In the current manuscript, they discuss the “The proportion of domain wall occupied surface before magnetization” and its correlation with OER activity changes. Reviewer 4 pointed out that the surface of the Ni/Fe alloy will have a different magnetization from the bulk.

a. I do not agree with the authors' response that the surface layer with different magnetization is thinner than the “active surface layer” such that this surface magnetization-effect should not play a role.

b. I think the situation is more complex:

In this schematic, I took the “bulk” picture from the manuscript and added the two separate surface layers. The key question arising from the debate between the referee and the authors and from the authors' previous work is: what will be the spin pinning in the (Ni/Fe)OxHy surface layer

2. Role of spin order: I agree with the authors that previous theoretical work (as listed in the response) provided a baseline for the argument that “material A with long-range FM order” has a higher OER activity than the same “material A without long-range FM order” due to changes in the reaction energetics and charge transfer kinetics which are influenced by the quantum spin exchange interactions (QSEI).

I find it very relevant to combine this nanoscopic picture with the mesoscopic picture including the domain walls. Considering the area of the surface that can be considered as “material A with long-range FM order” makes sense to me.

However, I agree with reviewer 4 that the surface magnetization deserves closer attention in the manuscript. I invite the authors to include such a consideration in the current manuscript, essentially combining their current work, the reviewer 4 input, and their previous work to arrive at a more complete picture.

3. I would be interested to learn what this means for the enhancement mechanism in the authors’ and the editors opinion. Is the OER, which is located at the surface, influenced by the magnetic field of the underlying domains? Or does the spin pinning on top of the domain walls change when magnetizing the samples, leading to a change in magnetic order in the active surface layer?

Response: We thank the reviewer for further comments and raising these discussions (points#1-3). We should first clarify that magnetization is the magnetic moment of the entire sample pulled in the direction of the magnetic field, not surface magnetization. Second, we fully agree with the micromagnetic simulations provided by Reviewer 4 that the magnetic domain structure of the surface is more complex than bulk. We here point out that the surface of the micromagnetic simulations is only 4 nm, and in the actual OER, the reactants are accessible to the bulk. Third, we thank the reviewer for the model diagram, and we agree that the model is more complex considering the magnetic domain structure of the surface. In fact, we have explained in the previous round in our response to Reviewer 4. The arrangement of magnetic moments on the surface and bulk are directly correlated, that is, the periodic arrangement of the magnetic domains and domain walls directly determines the magnetic configuration of the surface (Figure R1). When the magnetic field is applied, the average rate of change of surface magnetic moment is the same trend as the percentage of the magnetic domain wall in bulk. Thus, there is no problem to correlate OER enhancement using the occupation of magnetic domain walls.

Figure R1. a. The difference between two adjacent magnetic moments at different positions in the X-direction on the surfaces of samples with different thickness; **b.** The average rate of change of magnetic moment over a full stripe period.

I think these additions will make the paper much stronger and help resolve the debate between the authors and referee 4.

Response: We fully agree with the reviewer to add related discussion. The above data have been added into SI as Supplementary Fig.28. We have added the corresponding discussion at Line 290, Page 28 in SI:

“We cation that the domain structure in surface of those NiFe films is more complex than bulk, where the magnetic moment has a mostly in-plane arrangement with a continuous rotation. But, the depth of this surface is about 4 nm based on micromagnetic simulations single cell size, which is much less deep than the depth of the catalyst film surface involved in the reaction (at least 11 nm according to TEM). More importantly, the OER enhancement is from the magnetization, the important information for comparison comes from the “ Δ ”. The rate of change of the magnetic moment along the x-direction of the micromagnetic simulations surface shown in Supplementary Fig.28. The variation of magnetic moment is small in the domain region and large in the domain wall region. The rate of change of the magnetic moment at the surface corresponding to the domain wall shows a clear thickness dependence, i.e., the thicker the film and the wider the stripe, the smaller the rate of change of the magnetic moment. When the magnetic field is applied, all magnetic moment is pulled parallel, the thinner the sample, the greater the change in magnetic moment, which is consistent with the trend of OER enhancement.

Reviewer #4 (Remarks to the Author):

I thank the authors for their detailed and respectful reply to my comments. As stated in my first review, my background is in magnetism and I viewed the manuscript through this lens, which required additional clarifications to interpret the authors' results and claims correctly.

Given all information presented to me by the authors in their rebuttal, I can agree to the publication of the manuscript if the authors can clarify one final question and add a few minor things to the manuscript to improve its readability for the broad readership of the journal.

Response: Thanks for valuable suggestions from reviewer. We agree with the reviewer and have made the additional note as advised to improve the clarity and readability of this manuscript. We would like to thank the reviewer again to help improve the manuscript from a very professional insight of magnetism physics.

- Please add a small discussion on the fact that what is considered the "surface" for these reactions can be up to tens of nanometers deep into the material (as is indeed shown in the provided references) and place this in the context of the observed magnetization state (the MFM shows the magnetostatic fields "leaking through" from the bulk, whereas the surface (as defined as the monolayer thick boundary of the magnet), has a mostly in-plane arrangement that shows a continuous rotation). With this new information, it is now clear to me that my distinction between bulk and surface was not in line with how the surface should be considered for this reaction.

Response: Thanks for the reviewer's suggestion. We have added a small discussion about surface domain structure and the minimum depth of the catalyst film surface involved in the reaction in revised manuscript (marked in red).

Line 290, Page 28 in SI:

“We cation that the domain structure in surface of those NiFe films is more complex than bulk, where the magnetic moment has a mostly in-plane arrangement with a continuous rotation. But, the depth of this surface is about 4 nm based on micromagnetic simulations single cell size, which is much less deep than the depth of the catalyst film surface involved in the reaction (at least 11 nm according to TEM). More importantly, the OER enhancement is from the magnetization, the important information for comparison comes from the “ Δ ”. The rate of change of the magnetic moment along the x-direction of the micromagnetic simulations surface shown in Supplementary Fig.28. The variation of magnetic moment is small in the domain region and large in the domain wall region. The rate of change of the magnetic moment at the surface corresponding to the domain wall shows a clear thickness dependence, i.e., the thicker the film and the wider the stripe, the smaller the rate of change of the magnetic moment. When the magnetic field is applied, all magnetic moment is pulled parallel, the thinner the sample, the greater the change in magnetic moment, which is consistent with the trend of OER enhancement.

Supplementary Fig.28. a. The difference between two adjacent magnetic moments at different positions in the X-direction on the surfaces of samples with different thickness; **b.** The average rate of change of magnetic moment over a full stripe period.”

On a related, but important note, I agree that the presented data shows that the OER enhancement is related to the size of the domain walls in the samples. However, I still don't fully understand how these domain walls negatively affect the OER. The magnetization of the domain walls is no different than it is in the domains, it just lies in a different, gradually changing, direction. One explanation I can think of, given the clarification above, is that this reaction somehow probes a surface that is not only a few tens of nm deep, but also has such a large lateral dimension. On such length scales, the magnetization in the domain wall regions indeed cancels out (to a certain extent), which could explain a reduced local magnetization and hence the experimental data. Can the authors confirm that this is also their understanding (and if not, please provide an alternative explanation) and add this to the manuscript?

Response: Thanks for the comments. Based on the current understanding, we agree the explanation from the reviewer. The OER process typically occurs via four elementary steps involving different reaction intermediates and the formation of an O–O bond, which is eventually released as molecular oxygen. The two primary pathways proposed for the O–O bond formation are the water nucleophilic attack (WNA) and the interaction of two metal-oxo entities (I2M). In the case of WNA, there is stronger the 3d-2p hybridization and a higher spin density on the oxygen atoms (FM ligand holes) after spin alignment based on the previous DFT calculation (Nat Communications 12, 2608 (2021); Phys. Chem. Chem. Phys. 21, 2977–2983 (2019).), which will facilitate spin-selected charge transport and optimize the kinetics of the spin-charge transfer. However, the spins are in a disordered state in the domain wall region of NiFe film catalyst, which will not favor spin-polarized OER. The magnetic moment in the domain region is ordered, where the spin-polarized OER with low kinetic barrier is likely to occur. While the magnetic moment in the domain wall region is disordered, and no spin-polarized OER occurs in this region. In the I2M mechanism, the O–O bond formation involves the coupling of two separate metal-oxo moieties. In domain region, the spin in separate metal-oxo moieties are aligned, the ligand oxygens also

become polarized and the oxygen radicals are in parallel spin direction. The two unpaired electrons in parallel alignment are stabilized in oxygen antibonding π^* orbitals. The O-O coupling and triplet O_2 turnover can finally complete with a lower barrier. In domain wall region, in two nearest M-O \cdot radicals are not aligned, the ligand oxygens are unpolarized, where the spin polarization OER will not occur easily.

The scale range of this reaction mechanism includes both vertical and lateral dimension. In particular, the scale range of lateral dimension is much larger than domain.

- Finally, please add a small note warning the readers that the results of figure 4 should not be extrapolated beyond the shown regime.

Response: Thanks for the comments. We have revised in manuscript marked in red.

Line 224, Page 11 in main manuscript:

“(it should be noted that such correlation may not be extrapolated beyond the shown regime.)”